# Power Molding Inductors Prepared Using Amorphous FeSiCrB Alloy Powder, Carbonyl Iron Powder, and Silicone Resin

**DOI:** 10.3390/ma15103681

**Published:** 2022-05-20

**Authors:** Hsing-I Hsiang, Liang-Chih Wu, Chih-Cheng Chen, Wen-Hsi Lee

**Affiliations:** 1Department of Resources Engineering, National Cheng Kung University, Tainan 70101, Taiwan; e44071071@gs.ncku.edu.tw; 2Department of Mechanical Engineering, Far East University, Tainan 74448, Taiwan; ccchen@mail.feu.edu.tw; 3Department of Electrical Engineering, National Cheng Kung University, Tainan 70101, Taiwan; leewen@mail.ncku.edu.tw

**Keywords:** molding inductors, amorphous alloy powder, silicone resin, carbonyl iron powder

## Abstract

In this study, amorphous FeSiCrB alloy powder, carbonyl iron powder, and high-temperature heat-resistant silicone resin were used to prepare power molding inductors, and the effects of different heat treatment procedures on the magnetic properties were investigated. Two heat treatment procedures were used. Procedure 1: Amorphous FeSiCrB alloy powder was pre-heat-treated, then mixed with carbonyl iron powder and silicone resin and uniaxially pressed to prepare power inductors. Procedure 2: A mixture of amorphous FeSiCrB alloy powder, carbonyl iron powder, and silicone resin was uniaxially pressed. After dry pressing, the compacted body was heat-treated at 500 °C. Heat treatment after compaction can reduce the internal strain caused by high-pressure compaction and promote the crystallization of superparamagnetic nano-grains simultaneously. Therefore, the compacted sample after heat treatment exhibited better magnetic properties.

## 1. Introduction

Power molding inductors are primarily used in power modules for electromagnetic storage or noise filtering. They combine iron-based metal powder with coil winding to form a monolithic structure. In the case of power modules used in automobiles, the power module is frequently operated at a high frequency and in a harsh environment (high temperature and humidity). If the power components fail over time in a high-temperature environment, this will have a serious impact on driving safety. Therefore, a stable power supply for the electronic control unit is a major issue for automobile manufacturers. The operating voltage of the integrated circuit continues to fall, resulting in a significant increase in operating current. Power inductors used in DC power converters must be able to withstand high currents. When the operating current increases, the power inductors fail easily due to the temperature rise and inductance drop caused by magnetic saturation [1]. The power inductor’s temperature rise at an operating current is primarily determined by the DC resistance (DCR) and the power loss [2,3]. The coil length or number of turns used in a power inductor determines its DCR. When using a magnetic material with lower permeability, the coil length must be increased to achieve the required inductance, which raises the DCR. High DCR has a direct effect on I^2^R power losses and voltage drop because a high DC current flows through the power inductors. As a result, power inductors must have a lower DC resistance than general-purpose inductors. Molding power inductors’ power loss primarily comprises hysteresis loss and eddy current [4]. Magnetic core loss is dominated by hysteresis loss at low frequencies. Hysteresis loss is strongly influenced by the alloy’s coercivity and compaction density [5]. Eddy current loss increases rapidly with increasing frequency. Eddy current loss can be divided into two types: inter-particle and intra-particle eddy current loss [6]. Intra-particle eddy current loss is mainly controlled by the resistivity and particle size of the alloy powders [7]. The surface insulation coating on the powders, on the other hand, can reduce inter-particle eddy current loss [8].

As the DC current flowing through the power inductor exceeds the rated current, the inductance decreases significantly, lowering the DC–DC converter’s efficiency. Therefore, the inductor must ensure that the inductance does not drop too low under DC bias superposition. As a result, the electromagnetic properties of power inductors (low DCR, low power loss, superior DC-bias superposition characteristics, etc.) to meet the requirements of DC power converters has emerged as an important research topic [2].

Nano-crystalline and amorphous magnetic alloy materials have higher permeability, lower coercivity, and lower power loss than traditional polycrystalline magnetic alloys [9]. Because of their excellent soft magnetic properties, amorphous and nanocrystalline Fe-based alloys have become a popular magnetic material for a new generation of power molding inductors used in high-frequency electronic applications [10]. Magnetic nanocrystalline alloys, known as FINEMET, VITROPERM, or NANOPERM, are new soft magnetic alloys developed after amorphous alloys [11]. The amorphous alloy is at a metastable state. 

Amorphous alloys become stable through the crystallization process after heat treatment above the crystallization temperature, resulting in the formation of nanocrystalline alloys. The grain size of the nano-sized precipitates determines the soft magnetic properties of Fe-based nanocrystalline alloys. Superior magnetic properties (low magnetostriction, low power loss, and high permeability) of Fe-based nanocrystalline alloys are obtained when the crystallite size is smaller than the ferromagnetic (FM) exchange-related length (usually tens of nanometers) [12].

Molding power inductors are made by the uniaxial compaction of a mixture of magnetic alloy powder and organic resin. Internal strain in magnetic powders is generated by high molding compaction, resulting in decreased permeability, coercivity, and saturation magnetization [13]. The amorphous alloy must be annealed at high temperatures to reduce internal strain and promote the precipitation of superparamagnetic nanocrystalline grains [14]. However, most of the organic resins (epoxy and phenolic resins) used for power inductor molding decompose at temperatures above 400 °C, limiting the annealing process [15]. The silicone resin exhibits excellent thermal stability, hydrophobicity, adhesion, weather resistance, electrical insulation, and low chemical toxicity. In particular, the excellent thermal stability of silicone resin, which can reach temperatures above 600 °C [16], makes it an ideal annealing process for manufacturing nanocrystalline power molding inductors. Amorphous Fe-based alloy powders have high brittleness and hardness, making full density compacting difficult. Carbonyl iron powder (CIP) has a number of advantages, including a low cost and high saturation magnetization [17]. Hsiang et al. [18] reported that by incorporating CIP into FeSiCr alloy powders, the compaction density of the molding coils could be increased due to the large plastic deformation during pressing, and thus the inductance could be increased. In this study, the silicone resin was used as the organic binder for molding power inductors. The effects of the mixing ratio of carbonyl iron powder and different heat treatment procedures on the magnetic properties of nanocrystalline molding power inductors were investigated.

## 2. Experimental Method

The raw materials for this study were amorphous FeSiCrB alloy powder (Kuamet6B2, Epson Atmix Co., Ltd., Hachinohe, Japan) and carbonyl iron powder (Chung Yo Materials Co., Ltd., Kaohsiung City, Taiwan), as shown in Figure 1. The average particle sizes of nearly spherical amorphous FeSiCrB alloy powder and carbonyl iron powder are 50 μm and 3 μm. The silicone resin (RSN-0805, Dow Corning) was used as the binder. Two heat treatment procedures were used in this study. Procedure 1: To prepare the toroidal bodies, amorphous FeSiCrB alloy powder was pre-heated at 500 °C for 1 h in an atmosphere of 95% Ar + 5% H_2_ before being mixed with carbonyl iron powder and silicone resin and uniaxially pressed. Procedure 2: An amorphous FeSiCrB alloy powder, carbonyl iron powder, and silicone resin mixture was uniaxially pressed. Following dry pressing, the compacted body was heat-treated for 20 min at 500 °C in an atmosphere of 95% Ar + 5% H_2_. The magnetic powders with different mixing weight ratios of the amorphous FeSiCrB alloy powder to carbonyl iron powder (10:0, 7:3, 5:5, 3:7, and 0:10) were mixed with 2.5 wt% silicone resin for 1 h in a planetary mixer with acetone (99.5%, Sigma-Aldrich, St. Louis, MO, USA) as the solvent. The granulated powders were dry-pressed at 900 MPa into toroidal bodies with an outer diameter of 10 mm and an inner diameter of 5 mm after drying and granulation through a 40 mesh screen.

The samples were named nKmC-T, which represents a mixture of 10 n wt% amorphous FeSiCrB alloy powder and 10 m wt% carbonyl iron powder with different heat-treatment procedures (T: W (without heat treatment), B (heat treatment of amorphous FeSiCrB alloy powder at 500 °C for 20 min before compaction), and A (heat treatment at 500 °C for 20 min after compaction); for example, 3K7C-A: 30% amorphous FeSiCrB amorphous alloy powder and 70% carbonyl iron powder, which is heat-treated at 500 °C for 20 min after compaction. The chemical compositions and heat-treatment procedures of the samples are summarized in Table 1.

An X-ray diffractometer (Dandong Fangyuan, DX-2700, Dandong, China) with Cu Kα radiation was used to characterize the crystalline phases. SEM (Hitachi S4100, Tokyo, Japan) was used to examine the amorphous FeSiCrB alloy powder, carbonyl iron powder, and compact microstructures. The thermal behavior was investigated using a differential thermal analyzer and thermogravimetry (DTA/TG) (Netzsch STA 409C, Burlington, MA, USA). To obtain the saturation magnetization, the hysteresis loop was determined using a superconducting quantum interference device magnetometer (MPMS SQUID VSM, Quantum Design, San Diego, CA, USA). Initial permeabilities of the toroidal bodies were measured using an LCR meter (YHP 4291B, YHP Co., Ltd. Hyogo, Japan) with the HP 16454A magnetic material test fixture from 1 MHz to 1 GHz. A B-H analyzer was used to calculate the coercivities and power losses of toroidal bodies (SY-8218, IWATSU Electronics CO., Ltd., Tokyo, Japan). At 1 MHz, the DC-bias superposition characteristics of toroidal bodies were investigated using a magnetic device analyzer (WK3260B, Wayne Kerr Electronics Co., Ltd., London, UK) equipped with a DC bias current source (WK3265B, Wayne Kerr Electronics Co., Ltd., London, UK).

## 3. Results and Discussion

The DTA of the amorphous FeSiCrB alloy powder in a flowing N_2_ atmosphere is shown in Figure 2. The DTA curve clearly shows an exothermic peak at 540 °C, indicating that the powder was converted into crystalline. Instead of α-Fe, the crystalline phase may be a complicated solid solution of α-Fe (Si, Cr) due to short-range diffusion [19]. The XRD patterns of the amorphous FeSiCrB alloy powders before and after heat treatment are shown in Figure 3. For the amorphous FeSiCrB alloy powder, a significant broadening of the XRD peak was observed for the amorphous material. After 500 °C heat treatment, there was an increase in peak intensity and a decrease in full width at half maximum, associated with an increase in crystallinity due to crystallization. After heat treatment, the crystallite size calculated using the Scherrer equation was 20–30 nm. SQUID measurements for amorphous FeSiCrB alloy powders before and after heat treatment are shown in Figure 4. It indicates that heat treatment significantly increased saturation magnetization. The saturation magnetizations of the powders are 165 emu/g without heat treatment and 180 emu/g after heat treatment, respectively. Liu et al. [20] investigated the effect of annealing temperature on the magnetic properties of iron-based amorphous alloys and reported that the increase in saturation magnetization after annealing at 783 K was due to an increase in the crystalline volume fraction of the alloy, which led to a decrease in the intergranular distance and an enhancement of the coupling effect.

Figure 5 shows the variation in compaction densities of compacted bodies after heat treatment with various powder mixing ratios. Because of their high brittleness and hardness, the pure amorphous Fe-based alloy powder 10K0C-A had the lowest relative density. Compaction density can be increased by mixing carbonyl iron powder with amorphous FeSiCrB alloy powder. The relative density of the 5K5C-A sample is the highest. This is because the small carbonyl iron powder can fill the interstices of the large amorphous FeSiCrB alloy powder, reducing the porosity and increasing the relative density, as shown in Figure 6. As the amount of carbonyl iron powder added increased, the relative density decreased. This is due to carbonyl iron powder exceeding the amorphous FeSiCrB alloy powder’s interstice volume and widening interparticle distances, and the large amorphous FeSiCrB alloy powder cannot fill the small interstices formed by the small-sized carbonyl iron powders [18]. Due to the large plastic deformation of carbonyl iron powders during compaction, 0K10C-A has a higher relative density than 10K0C-A.

The effect of different powder mixing ratios and heat treatment procedures on the initial permeability is shown in Figure 7. Except for the 3K7C sample, the initial permeabilities of the heat-treated samples after compaction were significantly higher than those of the heat-treated samples before compaction. This is because of the high internal strain caused by the severe plastic deformation of carbonyl iron powders during uniaxial molding [21]. According to previous research [22], the high permeability of nanocrystalline alloy has been linked to two causes: (1) the magnetostriction of the alloy, which is proportional to the Si content, decreases dramatically after crystallization; (2) the effective magnetic anisotropy decreases dramatically for the very small grain size of α-Fe crystals due to ferromagnetic exchange interactions. Heat treatment reduced the internal strain after compaction while also promoting the crystallization of nanocrystalline grains in the amorphous matrix, resulting in a higher initial permeability. The annealing process was carried out in an atmosphere of 95% Ar + 5% H_2_ to avoid oxidation. However, carbonyl iron powder without phosphatizing was prone to rusting [23]. 3K7C-A contains a significant amount of carbonyl iron powder (70%), and the surface of the carbonyl iron powder may be oxidized after post-heat treatment, reducing the initial permeability. However, the initial permeability of 3K7C-B is higher than that of 3K7C-W, which has not been heat-treated. This is because, after heat-treating amorphous FeSiCrB alloy powder, a large amount of nanocrystalline α-Fe crystallized, improving the initial permeability. Because of its lowest compaction density, the pure amorphous FeSiCrB alloy 10K0C had the lowest initial permeability. Due to the highest relative density and crystallization of nanocrystalline α-Fe, 5K5C-A had the highest initial permeability, around 42.

The saturation magnetization is primarily determined by the compacted body’s relative density and constituents. The effect of the heat-treatment procedure on the saturation magnetization of 5K5C is shown in Figure 8, indicating that 5K5C-A had a higher saturation magnetization. This is because post-compaction heat treatment can eliminate internal strain caused by severe plastic deformation during uniaxial compaction.

Figure 9 shows the effect of the heat-treatment procedure on the power losses of toroidal bodies. It was found that power losses increased rapidly with frequency for all samples, primarily due to eddy current losses, and that heat treatment before and after compaction can reduce power loss, particularly in the high-frequency range. This is due to the crystallization of nanocrystalline α-Fe embedded in the residual amorphous matrix following heat treatment, which reduces coercivity and hysteresis loss. Because carbonyl iron powder has a lower electrical resistivity than amorphous nanocrystalline FeSiCrB alloy, increasing the carbonyl iron powder content should increase eddy current loss (power loss at high frequencies such as 2 MHz). The experiments, on the other hand, discovered that, when compared to the 10K0C-A, the addition of 30–70% carbonyl iron powder (7K3C-A, 5K5C-A, and 3K7C-A) can significantly reduce the power loss at 2 MHz from 580 kW/m^3^ to 300–400 kW/m^3^. The eddy current loss is made up of intra-particle and inter-particle eddy current losses [6]. According to Equation (1), the reduction in power loss at 2 MHz caused by the addition of carbonyl iron powder may be due to the smaller carbonyl iron powder particle size, which results in lower intra-particle eddy current loss [24].
(1)$$P_{e}=\frac{{\left(\pi d_{eff}B_{m}\right)}^{2}}{\beta \rho_{R}}f^{2}$$
where $P_{e}$: eddy current loss, $B_{m}$: saturation magnetization, *d_eff_*: effective particle size, *ρ_R_*: specific resistivity, and *β*: geometrical factor.

The temperature rise caused by power loss determines the size and operating frequency of the power inductors. By adding fine spherical carbonyl iron powder, the power losses of the amorphous FeSiCrB alloy at high frequency can be effectively reduced, increasing the operating frequency and reducing the size of the molding power inductors used in converters.

## 4. Conclusions

In this study, power molding inductors were prepared using amorphous FeSiCrB alloy powder, carbonyl iron powder, and high-temperature heat-resistant silicone resin, and the effects of different heat treatment procedures on the magnetic properties were investigated. Heat treatment after compaction reduces internal strain while also promoting the crystallization of nanocrystalline grains in the amorphous matrix, resulting in an increase in initial permeability. Carbonyl iron powder added to amorphous FeSiCrB alloy powder can increase the compaction density. This is due to the fact that the small-sized carbonyl iron powder can fill the interstices of the large amorphous FeSiCrB alloy powder, resulting in a reduction in porosity and an increase in relative density. Because of its high relative density and nanocrystalline α-Fe crystallization, 5K5C-A had the highest initial permeability, around 42. Heat treatment prior to and after compaction can reduce power loss, particularly in the high-frequency range. This is due to the crystallization of nanocrystalline α-Fe embedded in the residual amorphous matrix following heat treatment, which reduces coercivity and hysteresis loss. The addition of 30–70% carbonyl iron powder (7K3C-A, 5K5C-A, and 3K7C-A) can significantly reduce the power loss at 2 MHz from 580 kW/m^3^ to 300–400 kW/m^3^.

## Figures and Tables

**Figure 1 materials-15-03681-f001:**
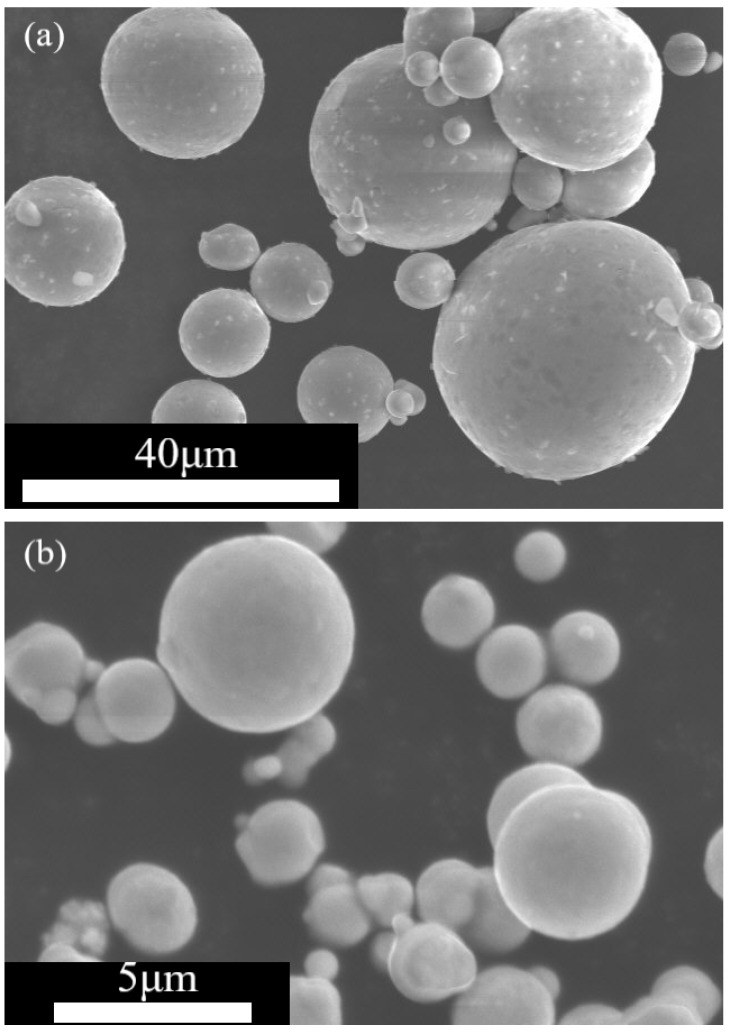
SEM images of (**a**) amorphous FeSiCrB alloy powder and (**b**) carbonyl iron powder.

**Figure 2 materials-15-03681-f002:**
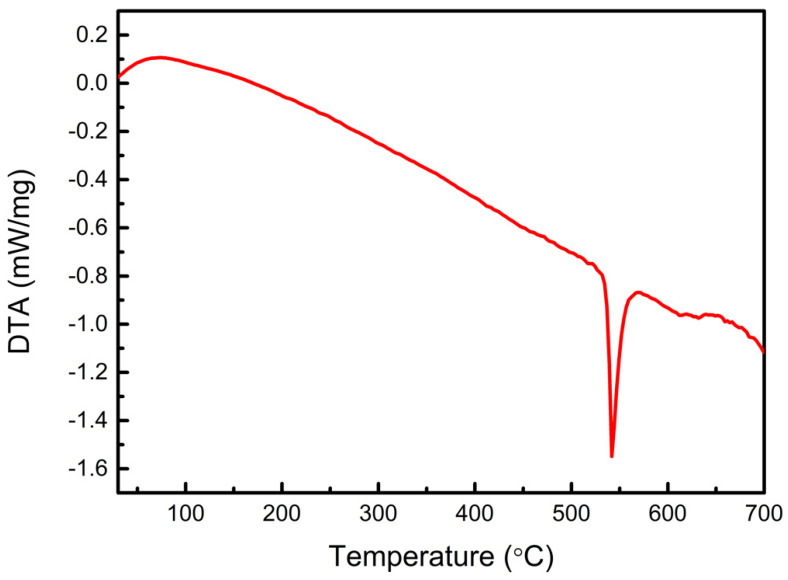
DTA result of the amorphous FeSiCrB alloy powder.

**Figure 3 materials-15-03681-f003:**
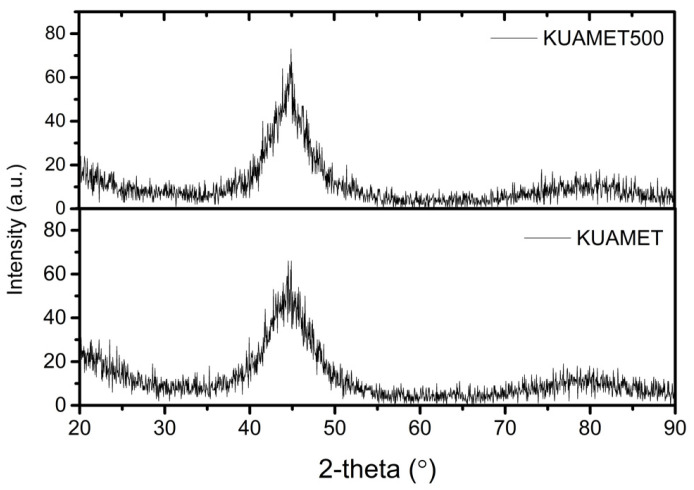
XRD patterns of the amorphous FeSiCrB alloy powders before (KUAMET) and after heat treatment (KUAMET 500).

**Figure 4 materials-15-03681-f004:**
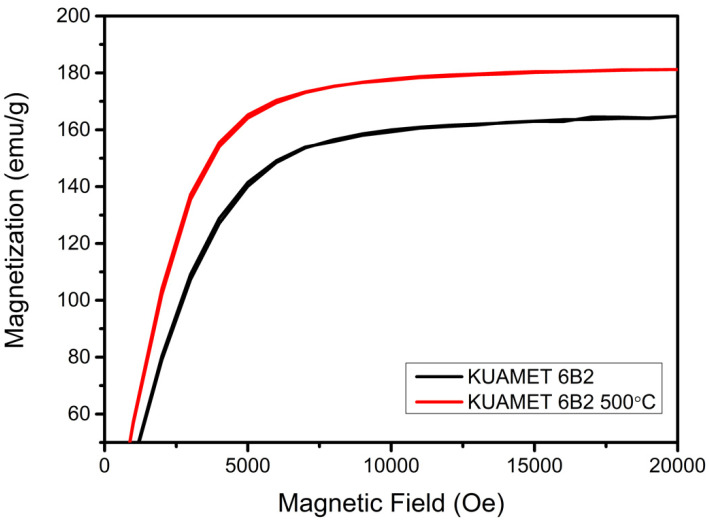
Magnetization versus magnetic field loops as measured by SQUID for amorphous FeSiCrB alloy powders before and after heat treatment.

**Figure 5 materials-15-03681-f005:**
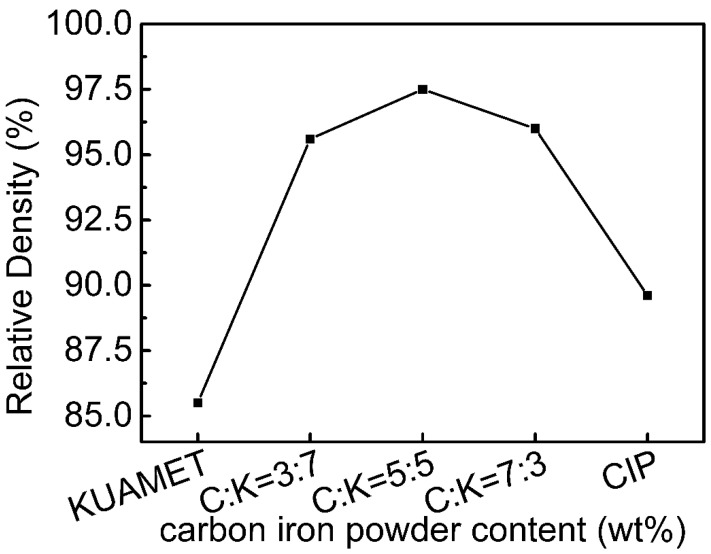
Variation of the compaction densities of the compacted bodies after heat treatment with different powder mixing ratios.

**Figure 6 materials-15-03681-f006:**
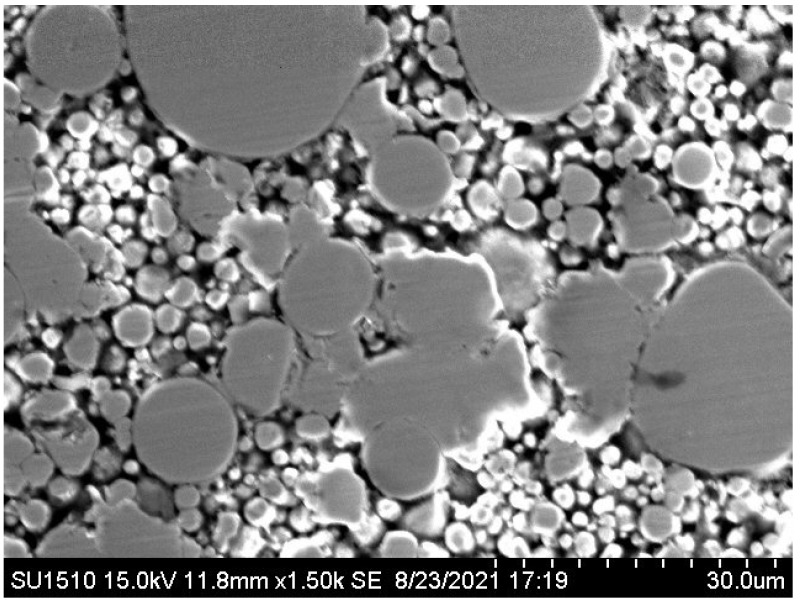
SEM image of the sample 5K5C-A.

**Figure 7 materials-15-03681-f007:**
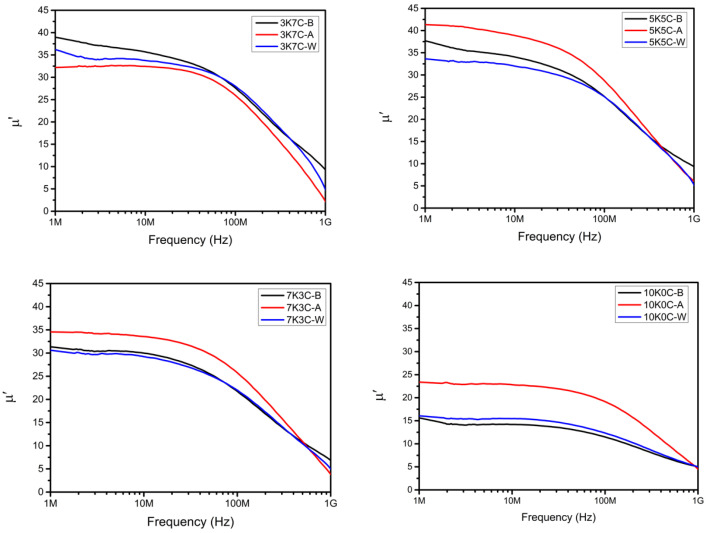
Influence of different powder mixing ratios and heat treatment procedures on the initial permeability.

**Figure 8 materials-15-03681-f008:**
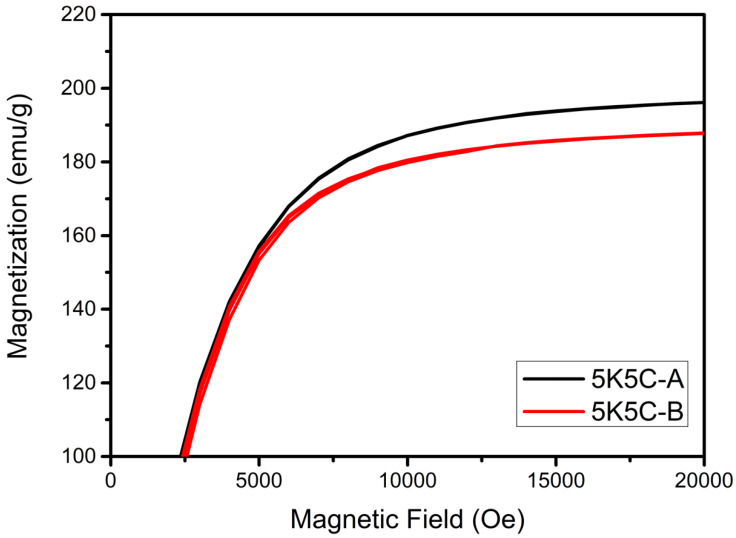
Effect of the heat-treatment procedure on the saturation magnetization of 5K5C (A: heat treatment after compaction; B: heat treatment before compaction).

**Figure 9 materials-15-03681-f009:**
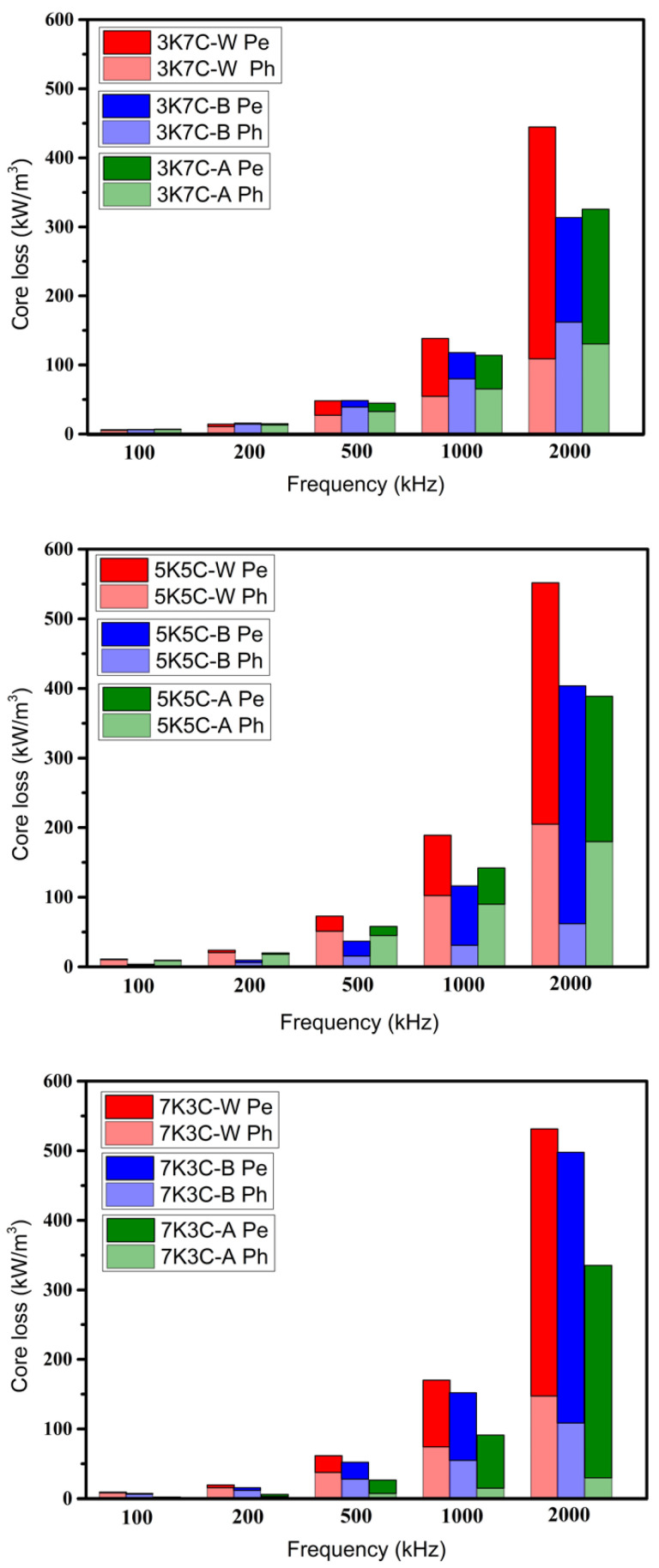
Effect of the heat-treatment procedure on the power losses of the toroidal bodies.

**Table 1 materials-15-03681-t001:** Chemical compositions and heat-treatment procedures of the samples.

Samples	Amorphous FeSiCrB Powder: Carbonyl Iron Powder	Heat Treatment of Amorphous FeSiCrB Powder at 500 °C	Silicone Resin
3K7C-W		Without heat treatment	
3K7C-B	3:7	Before compaction	2.5 wt%
3K7C-A		After compaction	
5K5C-W		Without heat treatment	
5K5C-B	5:5	Before compaction	
5K5C-A		After compaction	
7K3C-W		Without heat treatment	
7K3C-B	7:3	Before compaction	
7K3C-A		After compaction	
10K0C-W		Without heat treatment	
10K0C-B	10:0	Before compaction	
10K0C-A		After compaction	

## Data Availability

The data presented in this study are available in the article.

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
