# Peer review of "Power Molding Inductors Prepared Using Amorphous FeSiCrB Alloy Powder, Carbonyl Iron Powder, and Silicone Resin"

_materials, 2022, doi:10.3390/ma15103681_

Round 1
Reviewer 1 Report
I would suggest summarizing all the relevant information such as sample preparation details, notation, and, most important, numerical values of crucial electromagnetic parameters into a table. This would help a reader to grasp the essential content of the paper at a single sight.
Minor comments and some editorial suggestions have been inserted directly into the manuript file (attached).

Reviewer 2 Report
The authors report the study of two different methods to fabricate power moulding inductors, based on amorphous FeSiCrB alloy powder, carbonyl iron powder and high-temperature heat-resistant silicone resin. The comparison of the two method was established studying the effects on the magnetic properties.
The document needs of major revisions.
- line 99 “ acetone as the solvent” please add information of all chemicals (e.g. grade, provider, ...).
- the authors should add some comment of the role that play the oxygen during the treatment. In my opinion oxidation problem issues could be manifest. Please add some discussion about it.
-line 194 In the caption of figure 6,7, 8 the authors should add some comment about the samples colours to make easy the results reading. Or change the data name used in the legend ….It is difficult to understand the different material ratio or the used treatment only reading the name of samples.
- please add the errors bar in the figure 6,7.
Reviewer 3 Report
- Typo errors and grammatical mistakes should be carefully revised as I found some errors.
- In line 16, the sentence should be “heat-treated at 500°C”.
- Page 3 paragraph must re-written, It will create a confusion to readers. Because author has mentioned that, they mixed all the powders in planetary mixer for 1 hour, then pressed the same at 900MPa load to obtain toroidal shape. But in the heat treatment procedure 1, author has mentioned that powders were pre-heated. Once they pressed to toroidal shape, how do authors get powders again? Eventhough I understood what authors want to say, but it is confusing. Please revise it.
- Heat treatment procedure 1, what is the atmosphere used to pre-heat the FeSiCrB powders?
- Discussion for Fig. 2, 3 and 4 is not sufficient. It is recommended to discuss more about the results of DTA, XRD, and SQUID measurements.
- Why did author performed DTA analysis only for FeSiCrB powders? It is recommended to perform DTA analysis even for mixture of FeSiCrB alloy powder and carbonyl iron powder.
- The experiment on the power losses of the toroidal bodies looks good, but it is recommended to show the microstructure of the toroidal shape (pressed powders) using optical or electron microscopes.
Round 2
Reviewer 2 Report
in my opinion the paper is now ready to publish
Reviewer 3 Report
Author has revised the manuscript as per my suggestion. I am satisfied with the author responses to my comments